# The Discovery of Small ERK5 Inhibitors via Structure-Based Virtual Screening, Biological Evaluation and MD Simulations

**DOI:** 10.3390/molecules30214181

**Published:** 2025-10-25

**Authors:** Noor Atatreh, Radwa E. Mahgoub, Rose Ghemrawi, Molham Sakkal, Nour Sammani, Mostafa Khair, Mohammad A. Ghattas

**Affiliations:** 1College of Pharmacy, Al Ain University, Abu Dhabi 64141, United Arab Emiratesrose.ghemrawi@aau.ac.ae (R.G.);; 2AAU Health and Biomedical Research Centre, Al Ain University, Abu Dhabi 64141, United Arab Emirates; 3Core Technology Platforms, New York University Abu Dhabi, Abu Dhabi 129188, United Arab Emirates

**Keywords:** ERK5, lung cancer, drug discovery, inhibitor, virtual screening, anticancer agents

## Abstract

ERK5, a member of the MAP kinase family, has been implicated in several cancer types due to its role in regulating cell proliferation, survival, and migration. In this study, structure-based virtual screening was employed, followed by cell assays, and molecular dynamics simulations to identify novel ERK5 inhibitors. A commercially available library of 1.6 million compounds was subjected to a three-stage docking process (HTVS, SP, and XP), using the docking module in Schrodinger Maestro, yielding 40 candidates with superior docking scores compared to the co-crystallized ligand. These compounds were then tested for antiproliferative activity using an MTT assay in A549 and H292 lung cancer cell lines. Among the hits, STK038175, STK300222, and GR04 showed significant activity with IC_50_ values of ranging from 10 to 25 µM. Western blot analysis revealed that STK300222 at 50 µM reduced the phosphorylation of ERK5 downstream targets similarly to a known inhibitor, while wound healing assays confirmed a dose-dependent decrease in cell migration. Molecular dynamics simulations of 200 ns further demonstrated that all three compounds form stable complexes with ERK5 that are comparable to the co-crystallized ligand in 5BYZ. The MD simulations also revealed strong electrostatic and solvation interactions observed for STK300222 and GR04 particularly. Furthermore, by calculating the MM-GB/SA scores from the MD trajectories, the binding affinities of the three hits, along with the co-crystallized ligand in 5BYZ, were re-scored. Although the co-crystallized ligand had the highest MM-GB/SA score at −38.96 Kcal mol^−1^, STK300222 had a comparable score of −35.45 Kcal mol^−1^. These results highlight STK300222 and GR04 as promising candidates for further optimization and in vivo validation as ERK5 inhibitors.

## 1. Introduction

Cancer remains a major global health burden, ranking second only to cardiovascular disease as a leading cause of death worldwide. According to global health data from 2022, cancer accounted for nearly 10 million deaths and close to 20 million new diagnoses worldwide [1]. It is projected that roughly 20% of the global population will experience a cancer diagnosis in their lifetime. Furthermore, cancer-related mortality continues to be high, with men facing an estimated lifetime risk of one in nine and women one in twelve [1].

Among all cancer types, lung cancer stands out as the most commonly diagnosed in 2022 and accounted for nearly one in every eight new cancer cases globally [1]. It also remains the foremost cause of cancer-related mortality, contributing to approximately 18.7% of all cancer deaths [1]. Given its widespread global impact and significant mortality burden, cancer remains a central focus of ongoing research, with considerable efforts directed toward the development of new therapeutic strategies.

While several targeted therapies such as erlotinib, crizotinib, and cabozantinib have demonstrated clinical efficacy in the treatment of lung cancer, their use is often accompanied by significant adverse effects [2,3]. Moreover, the frequent emergence of drug resistance further limits their long-term effectiveness [2,3]. These challenges highlight the pressing demand for innovative anti-lung-cancer therapies that offer enhanced safety and sustained therapeutic efficacy.

Extracellular signal-regulated kinase 5 (ERK5), or Big MAP Kinase-1 (BMK-1), is a component of the mitogen-activated protein kinase (MAPK) family [4]. It plays a crucial role in controlling essential cellular functions, including differentiation, proliferation, survival, migration, and tumour development [5]. ERK5, produced by the MAPK7 gene, belongs to the family of serine/threonine kinases. Its elevated expression has been consistently linked to poor prognosis across a range of malignancies, suggesting its value as a prognostic biomarker. In particular, increased ERK5 expression is frequently observed in human lung cancer, where it is thought to contribute to tumour progression. High levels of ERK5 have also been associated with cancer hallmarks such as epithelial–mesenchymal transition, therapeutic resistance, diminished patient survival, and heightened invasive and metastatic capacity [6,7,8]. Altogether, these observations highlight ERK5 as a therapeutically relevant target, especially in lung cancer, where its dysregulation may present a strategic opportunity for targeted intervention.

ERK5 signalling influences several critical downstream targets, including Phospho-c-Fos [5], Phospho-CREB [4], and Phospho-Fra1, which collectively regulate essential cellular processes. Phospho-c-Fos, a component of the AP-1 transcription factor complex, drives gene expression programs associated with cell proliferation and differentiation. Phospho-CREB promotes survival signalling and confers resistance to apoptosis. In parallel, activated Fra1 is involved in promoting tumour cell invasiveness and metastatic behaviour, in part through its regulatory influence on epithelial–mesenchymal transition (EMT) and matrix metalloproteinase (MMP) activity.

In contrast to the closely related kinases ERK1 and ERK2, ERK5 contains an extended C-terminal domain with transcriptional activation capabilities, which amplifies its role in cellular signalling, particularly under mitogenic stimulation and cellular stress [4,9]. This distinct structure underscores not only the functional uniqueness of ERK5 but also its emerging significance as a molecular target, especially within the field of oncology.

Structurally, ERK5 consists of a canonical kinase domain at the N-terminus followed by a lengthy C-terminal, which includes the transcriptional activation domain [4,9]. This configuration is critical not only for its enzymatic activity but also for its role in nuclear signalling, making targeting it a promising strategy for inhibitor design. The kinase domain of ERK5 shares common features with other MAP kinases, which includes the ATP-binding site, which has been the focus for many small molecule inhibitors [4,10].

ERK5’s activation is linked to the development and progression of a number of malignancies, primarily through its ability to drive key transcriptional programs that promote cell division and survival [10]. This has rendered targeting ERK5 an attractive strategy for cancer therapy, particularly in malignancies that result in elevated levels, or hyper activation. Furthermore, ERK5 is involved in anti-apoptotic signalling and angiogenesis, processes integral to cancer cell survival and tumour growth, thus amplifying its attractiveness as a therapeutic target.

While the search for ERK5 inhibitors is ongoing, with novel inhibitors being reported, efforts utilizing computer-aided drug design, particularly virtual screening protocols, are not abundant in the literature. One paper details a rather intricate workflow, using a number of elegant computational tools, including virtual screening utilized to find potentially active flavonoids that can act as inhibitors of ERK5 [11]. They make use of other tools like docking, MM-GB/SA rescoring, and density functional theory (DFT) for providing a preliminary pIC_50_ for the most promising hits. Other papers studying ERK5 have used docking tools, to provide evidence of their hits, showing favourable binding to the ATP-binding cavity. Other studies also aimed to analyse the different possible binding confirmations of their hits, retrieved via more classical methods, such as synthesis [12,13,14].

Despite its therapeutic potential, targeting ERK5 has proven challenging. The most well-characterized ERK5 inhibitors, such as XMD8-92 [15] and AX15836 [16], have demonstrated significant efficacy in inhibiting ERK5 activity both in vitro and in various cancer models. However, these inhibitors often suffer from common drawbacks, including poor selectivity, potentially resulting in non-specific interactions with other targets and associated toxic responses [17,18].

This manuscript aims to find novel small lead-like inhibitors targeting ERK5, employing structure-based virtual screening complemented by biological evaluations and molecular dynamics simulations to offer a comprehensive view of their potential and limitations in cancer therapy.

## 2. Results and Discussion

### 2.1. Docking and Shortlisting of Hits

The virtual screening protocol used to filter the extensive compound database, comprising ~1.6 million ligands from TimTec and ChemDiv combined, is depicted in Figure 1, illustrating the filters applied to reduce the initial library to ~1.1 million ligands. By implementing drug-like rules (Lipinski’s and Veber’s), the selected ligands were ensured to adhere to established drug-like properties. Additionally, an online PAINS filter was employed to exclude potential promiscuous from the screened dataset. The filtered ligand library was sequentially docked into the ATP-binding site of ERK5 (PDB: 5BYZ). To optimize computational efficiency, a three-tier docking protocol was employed. Initially, all ligands underwent high-throughput virtual screening (GLIDE-HTVS), after which the top 20% were re-docked using standard precision (GLIDE-SP), followed by a final round of extra precision docking (GLIDE-XP), progressively increasing accuracy at the expense of computational cost.

To validate the docking protocol and set a selection threshold, the co-crystallized ligand from the ERK5 crystal structure (PDB ID: 5BYZ) was redocked into its native binding site, reporting a docking score of −9.26 Kcal mol^−1^. This value was used to define a filtering criterion, whereby compounds scoring better than −9.0 Kcal mol^−1^ were retained for further analysis. From this pool, the top 500 ligands were shortlisted. Each was visually examined within the ERK5 ATP-binding site to assess spatial complementarity and key interactions with surrounding amino acid residues. To exclude false positives associated with nonspecific binding, all candidates were subjected to aggregator screening using both Aggregator Advisor and the BAD Scam filter. Only those compounds that triggered no alerts were selected for biological evaluation, resulting in a final set of 21 compounds from TimTec and 19 from ChemDiv for cellular testing.

### 2.2. Screening of Potential ERK5 Inhibitors in A549 and H292 Lung Cancer Cell Lines

The A549 and H292 lung cancer cell lines were selected based on evidence demonstrating high expression of ERK5 and its critical involvement in promoting lung cancer cell proliferation, and migration [7,19,20]. Notably, ERK5 inhibition has been previously shown to suppress proliferation and impair tumour progression in lung cancer models [19]. Building on this, we screened the short-listed 40 compounds using the MTT assay to evaluate their potential in inhibiting proliferation in lung cancer cell lines (Appendix A). Among these, three compounds—STK038175, STK300222, and GR04 (Table 1)—demonstrated significant efficacy in reducing phosphorylation of downstream ERK5 target proteins (Phospho-c-Fos, Phospho-CREB, and Phospho-Fra1) in A549 and H292 cells. The IC_50_ values for each compound were determined through MTT assays, utilizing serial dilutions starting from 100 µM (Table 1). Based on these IC_50_ values, treatment concentrations were set at the IC_50_ and twice the IC_50_ (IC_50_ × 2) to assess dose-dependent effects.

Several studies have reported ERK5 inhibitors with markedly lower IC_50_ values compared to those listed in Table 1. For example, AX15836 was developed through the introduction of a methyl sulphonamide on the N-11 nitrogen, resulting in an inhibitor with improved selectivity over BRD4 and potent cellular activity (IC_50_ ≈ 86 nM). However, its poor solubility limited activity at higher concentrations, likely due to compound precipitation, and it showed no significant antiproliferative effect in an IL-6-dependent assay, suggesting possible non-ERK5-related mechanisms [21]. BAY-885 (IC_50_ ≈ 200 nM biochemically) and JWG-071 (IC_50_ ≈ 450 nM biochemically) were subsequently developed as tool compounds with greater selectivity, though both require micromolar concentrations in cell-based systems [22,23]. In addition, ADTL-EI1712 exhibited strong inhibition of ERK1 (IC_50_ = 40 nM) and ERK5 (IC_50_ = 65 nM) and suppressed tumour growth in xenograft models. Yet, it also induced massive cytoplasmic vacuolization in MKN-74 cells, indicating potential off-target pharmacology [21].

Together, these findings highlight that while more potent ERK5 inhibitors exist, their development and clinical translation are complicated by challenges related to solubility, selectivity, and off-target effects.

### 2.3. Validation of ERK5 Inhibition and Downstream Phosphorylation Reduction in Lung Cancer Cell Lines

Western blot analysis (Figure 2 and Appendix A) was conducted to evaluate the effects of the candidate ERK5 inhibitors—STK038175, STK300222, GR04—on the phosphorylation levels of three well-characterized ERK5 downstream targets: c-Fos, CREB, and Fra-1. These transcription factors were selected based on their established regulation by ERK5 signalling. Phosphorylation was detected using phospho-specific antibodies, with GAPDH serving as a loading control to ensure consistent protein normalization. To benchmark the efficacy of the novel compounds, the known ERK5 inhibitor XMD8-92 was included as a positive control at 5 µM—a concentration widely used in lung cancer models [19]. This enabled a direct and consistent comparison of signalling inhibition between the novel compounds and a validated ERK5 inhibitor.

In A549 cells, treatment with STK038175 significantly reduced the phosphorylation levels of all three targets (Phospho-c-Fos, Phospho-CREB, and Phospho-Fra1), with stronger inhibition observed at 20 µM compared to 10 µM (*p* < 0.01 for both concentrations vs. DMSO control). However, the extent of inhibition was consistently lower than that achieved by XMD8-92 (*p* < 0.01 for STK038175 vs. XMD8-92). STK300222 demonstrated modest inhibition across all three phospho-proteins at 25 µM (*p* < 0.05), but no further improvement was observed at 50 µM, resulting in significantly weaker effects than XMD8-92 at both doses (*p* < 0.001 for STK300222 vs. XMD8-92). GR04 achieved substantial suppression of Phospho-c-Fos, Phospho-CREB, and Phospho-Fra1 at both 10 µM and 20 µM, with near-complete inhibition at the higher dose (*p* < 0.001 for 20 µM vs. DMSO control). Notably, the inhibitory effects of GR04 at 20 µM were comparable to XMD8-92 for Phospho-CREB and Phospho-Fra1 (*p* > 0.05 vs. XMD8-92), although slightly less potent for Phospho-c-Fos (*p* < 0.01 vs. XMD8-92). The known ERK5 inhibitor XMD8-92 consistently abolished phosphorylation of all three proteins (*p* < 0.001), confirming its potency and serving as a benchmark for the novel compounds.

In H292 cells, the inhibitors showed reduced efficacy compared to A549 cells. STK038175 caused mild reductions in the phosphorylation of Phospho-c-Fos, Phospho-CREB, and Phospho-Fra1 at both 10 µM and 20 µM, with statistical significance only at 20 µM for Phospho-c-Fos and Phospho-CREB (*p* < 0.05 vs. DMSO control). However, the inhibitory effects of STK038175 were significantly weaker than XMD8-92 across all targets (*p* < 0.01 for STK038175 vs. XMD8-92). STK300222 exhibited moderate inhibition of the three phospho-proteins at 25 µM (*p* < 0.05), but no further reduction was observed at 50 µM, resulting in significantly lower efficacy compared to XMD8-92 at both doses (*p* < 0.001 for STK300222 vs. XMD8-92). GR04 demonstrated strong suppression of Phospho-c-Fos, Phospho-CREB, and Phospho-Fra1 at 10 µM and near-complete inhibition at 20 µM (*p* < 0.001 for 20 µM vs. DMSO control). The effects of GR04 at 20 µM were comparable to XMD8-92 for Phospho-CREB and Phospho-Fra1 (*p* > 0.05 vs. XMD8-92) but remained slightly less effective for Phospho-c-Fos (*p* < 0.01 vs. XMD8-92).

In conclusion, GR04 showed robust inhibition of these targets with efficacy comparable to XMD8-92 for Phospho-CREB and Phospho-Fra1, making it a promising ERK5 inhibitor candidate. STK300222 exhibited moderate inhibition, though its lack of dose-dependent improvement suggests limitations in potency or specificity. STK038175 demonstrated weaker effects across both cell lines, indicating suboptimal target engagement.

The stronger response observed in A549 cells may be explained by their KRAS-mutant status, which is known to increase reliance on the MAPK/ERK signalling pathway, including ERK5 [24]. In KRAS-mutant cancers, ERK5 often plays a key role in supporting cell growth, survival, and migration. On the other hand, H292 cells are KRAS wild-type and may use other signalling pathways to maintain these functions, making them less dependent on ERK5 and therefore less responsive to its inhibition [24]. This difference in pathway dependency likely accounts for the reduced sensitivity of H292 cells to the tested compounds. These results suggest that ERK5-targeted therapies may be more effective in tumours with specific genetic profiles, such as KRAS mutations.

### 2.4. Inhibition of Cell Migration by ERK5 Inhibitors in A549 and H292 Lung Cancer Cell Lines

The wound healing assay was conducted to assess the effects of ERK5 inhibitors on cell migration in A549 (Figure 3) and H292 (Figure 4) lung cancer cell lines. Cells were treated with DMSO as a control or various concentrations of STK038175 (10 µM and 20 µM), STK300222 (25 µM and 50 µM), and GR04 (10 µM and 20 µM). Wound closure was monitored at 0, 24, 48, and 72 h, and the percentage of wound closure was calculated relative to the initial wound size.

As shown in Figure 3, DMSO-treated A549 cells exhibited steady wound closure, reaching complete closure by 72 h. In contrast, treatment with STK300222 resulted in significant inhibition of wound closure at both concentrations, with the strongest effect observed at 50 µM, where wound closure remained minimal (~20%, *p* < 0.001 vs. DMSO). GR04 at 20 µM also effectively prevented wound closure, maintaining levels below 25% at 72 h, though STK300222 at 50 µM demonstrated slightly superior inhibition (*p* < 0.05). STK038175 exhibited a minor inhibitory effect at 10 µM, reducing wound closure to ~70% by 72 h (*p* < 0.05 vs. DMSO).

In H292 cells (Figure 4), untreated and DMSO-treated cells displayed progressive wound closure, reaching complete closure by 72 h. Similar to A549 cells, STK300222 exhibited strong inhibitory effects, particularly at 50 µM, where wound closure was limited to ~30% at 72 h (*p* < 0.001 vs. DMSO). GR04 at 20 µM also prevented wound closure, maintaining levels below 40% by 72 h (*p* < 0.01 vs. DMSO), though its effect was slightly less pronounced than STK300222 at 50 µM. STK038175 had a minor inhibitory effect at 20 µM, reducing wound closure to ~80% at 72 h (*p* < 0.05 vs. DMSO), with no significant effect observed at 10 µM.

Overall, the results highlight STK300222 at 50 µM as the most effective compound in suppressing cell migration, followed by GR04, with consistent performance across both A549 and H292 lung cancer cell lines. The superior efficacy of STK300222 and GR04 is further supported by Western blot analysis, which demonstrated significant inhibition of phosphorylation of key downstream ERK5 targets—Phospho-c-Fos, Phospho-CREB, and Phospho-Fra1. This alignment between molecular inhibition and functional anti-migratory effects emphasizes their potential as therapeutic ERK5 inhibitors.

The strong correlation between the suppression of ERK5 phosphorylation targets and reduced cell migration underscores the central role of ERK5 in driving cancer cell motility. These findings validate STK300222 and GR04, particularly at higher concentrations, as promising candidates for further investigation as therapeutic agents for the treatment of metastatic lung cancer. Taken together, the results position STK300222 and GR04 as promising candidates for further investigation, with future efforts focused on structural refinement to achieve stronger activity at lower doses.

Collectively, our findings demonstrate that the observed effects—namely, reduced phosphorylation of ERK5 downstream targets and impaired cell migration—are the result of specific ERK5 pathway inhibition rather than general cytotoxicity. All assays were conducted at sub-lethal concentrations (IC_50_ and IC_50_ × 2), where cell morphology and adherence were maintained. Importantly, the consistent and dose-dependent suppression of Phospho-c-Fos, Phospho-CREB, and Phospho-Fra1 support a target-specific mechanism. These results validate GR04 and STK300222 as promising ERK5 inhibitors and highlight the importance of ERK5 in regulating migration and signalling independently of cell viability loss.

### 2.5. MD and MM-GB/SA Rescoring

The docking methodology implemented in this study relied on a flexible ligand–rigid receptor approach, without incorporating the dynamic nature of the binding pocket. Although the top three candidates demonstrated notable anticancer potential in cellular assays, these results did not definitively establish direct engagement with ERK5. To further evaluate their binding stability and interaction patterns, molecular dynamics simulations extending over 200 ns were performed with each ligand initially placed in the ERK5 active site based on its docked pose. A parallel simulation was also conducted for the co-crystallized ligand extracted from PDB entry 5BYZ, serving as a reference for comparative analysis of binding strength and structural stability (Figure 5a). To monitor the integrity and persistence of the interactions, root-mean-square deviation (RMSD) calculated using ligand-to-ligand was plotted for both the ligands and the protein backbones throughout the simulations.

Our two promising hits, STK300222 and GR04 exhibited IC_50_ values at 25 µM and 10 µM, respectively. Upon comparing their RMSD plots with that of the co-crystallized ligand (Figure 5a), GR04 exhibits a fluctuation of around 1.75 Å, ranging from 0.5 to 2.25 Å throughout the entire simulation (Figure 5b). This is rather comparable to the co-crystallized ligand in 5BYZ, where the fluctuation of the ligand’s plot ranges is around 2 Å (Figure 5a), but its range is rather slightly higher, from 2.5 to 4.5 Å. This can be attributed to its relatively high molecular weight (465.5 Da) and substantial conformational flexibility, as indicated by its nine rotatable bonds. STK300222 on the other hand, experienced significant fluctuations, ranging from 0.5 to 3.5 Å during the initial 20 to ~80 ns of the simulation (Figure 5c). After this period, its RMSD profile stabilized somewhat, maintaining a narrower fluctuation range between 2.5 and 3.5 Å. This pattern suggests a conformational shift early in the simulation where STK300222 adjusted to find a more favourable binding pose within the binding site. Throughout the remainder of the simulation, STK300222 showed relative stability, making minimal adjustments to refine its fit within the active site. Upon clustering to elucidate the most representative pose for every 50 ns of the simulation, the ligand shows clear readjustment within the binding cavity, to achieve a more favourable fit. Throughout these dynamics, the protein backbone remained consistently stable in complex with both ligands, with RMSD values not exceeding 2.25 Å. STK038175 on the other hand exhibited the most stable ligand-protein complex among our top hits (Figure 5d). It displayed minimal fluctuations in the RMSD plot, maintaining a range between 0.25 and 1.25 Å throughout the 200 ns simulation. Likewise, the protein backbone remained largely stable throughout the simulation, with RMSD values fluctuating between ~1.0 and 2.5 Å. Notably, STK038175 was more stable compared to the co-crystallized ligand, which exhibited greater fluctuations in its RMSD plot, ranging from 2.5 to 4.5 Å. However, the backbone’s stability in both cases was similar, remaining within the range of ~1.0–2.5 Å.

Besides the original analysis calculating the ligand RMSD relative to the ligand itself (ligand-to-ligand RMSD), primarily reflecting the internal flexibility and conformational changes of the ligand throughout the trajectory, a ligand-to-pocket RMSD analysis was performed to better assess ligand retention within the binding site by aligning the trajectory on the backbone atoms of the binding pocket residues (Figure 6).

The ligand-to-pocket RMSD results reveal distinct behaviours among the four ligands. STK300222 exhibits the highest values, plateauing around 6–7 Å with a brief spike to 8 Å. 5BYZ, the co-crystallized ligand, shows values between 4 and 6.5 Å, while GR04 fluctuates between approximately 3 and 5.5 Å. STK038175 remains the most stable, with RMSD values around 2–3 Å. Visual inspection of the trajectories confirms that none of the ligands dissociate from the binding pocket during the 200 ns simulation. Clustering of the trajectories every 50 ns indicates that the higher RMSD values observed for 5BYZ and STK300222 arise from pose reorientation within the pocket, whereas GR04 shows moderate reorientation and STK038175 remains largely fixed.

Such elevated RMSD values in the absence of dissociation have been previously reported in multiple molecular dynamics studies. One study showed that a ligand bound to CDK2 (PDB 2B52) exhibited RMSD values exceeding 6 Å due to pose rearrangements within the pocket rather than unbinding [25]. Another observed similar behaviour for Abl–imatinib complexes, where high ligand RMSD values occurred despite the ligand remaining stably anchored [26].

Accordingly, our higher RMSDs—especially for 5BYZ and STK300222—are best explained by pose flips/reorientations in the binding pocket, rather than exit from the site.

Docking scores may not be the best tool for ranking of top-hits, as they are often biased towards bigger ligands with higher molecular weights, regardless of their fitting and binding modes [27]. Accordingly, binding free energy calculations using the MM-GB/SA method were performed on the ligand–protein complex following initial equilibration. Accordingly, it was calculated using the last 30 ns of the simulation (from 170 ns to 200 ns). In line with the cell-based assay results, STK300222 appears to have the best affinity towards ERK5, achieving an MM-GB/SA score close to that of the co-crystallized ligand, registering −35.45 Kcal mol^−1^ versus −38.96 Kcal mol^−1^, respectively. STK038175 comes in second, with a MM-GB/SA score of −28.76 Kcal mol^−1^. Due to the conformational change brought about by the ligand, GR04 reports the worst MM-GB/SA of −24.15 Kcal mol^−1^. Overall, STK300222 portrays the best stability and binding affinity towards ERK5, comparable to the co-crystallized ligand. The remaining two ligands also demonstrated acceptable stability and binding affinity; however, GR04 exhibited greater conformational adjustment within the active site over the course of the simulation.

The MM-GB/SA calculations indicated that STK300222 had the most favourable binding free energy among the three hits; however, this outcome does not fully align with the in vitro IC_50_ values, where STK300222 (25 µM) was less potent than STK038175 and GR04 (both 10 µM). Such discrepancies are expected, since IC_50_ values capture not only target binding but also additional cellular influences—including membrane permeability, efflux, and metabolic stability—that are not accounted for in MM-GB/SA scoring. Moreover, because MM-GB/SA was evaluated over the final 30 ns of the trajectory (post initial equilibration), the apparent stability of STK300222 during this period likely contributed to its more favourable predicted binding energy. Notably, when MM-GB/SA was instead assessed over the initial 30 ns, the rank order of binding affinities was more consistent with the IC_50_ values, suggesting that dynamic conformational states sampled earlier in the simulations may also play a role in shaping compound activity (Appendix A).

### 2.6. Binding of Top Hits and Pairwise Residue Analysis

Although running multiple shorter simulations (e.g., 3 × 65 ns replicas) can enhance conformational sampling by exploring alternative trajectory pathways, in this study we employed clustering analysis instead. Clustering has been widely used to extract biologically meaningful conformational states from single MD runs [28]. Here, clustering was applied to mitigate, at least in part, the limitations of relying on a single continuous trajectory, enabling us to identify and analyze the dominant conformations sampled at different time intervals (every 50 ns). While this conventional clustering approach may be less sensitive to rare transitional states, it is well suited to capture the major ligand-binding poses within the pocket.

STK038175 demonstrated a notably stable RMSD profile over a 200 ns MD simulation, showing minimal movement throughout the trajectory as evidenced by the clustering of MD trajectories (Figure 7). This stability is particularly significant considering that STK038175’s MM-GB/SA score was somewhat comparable to that of the co-crystallized ligand.

During the simulation, the co-crystallized ligand initially positioned itself vertically within the binding site, with its piperidinyl ethyl formamide side chain directed downwards, protruding from the active site (Figure 8a). Over the course of the simulation, the ligand underwent a notable conformational shift, reorienting its side chain into a horizontal alignment parallel to the active site surface (Figure 8b). This readjustment likely contributed to the fluctuations observed in the RMSD plot, indicating dynamic binding behaviour. Meanwhile, STK038175 portrayed only one binding pose throughout the 200 ns simulation, explaining its superior RMSD plot in terms of stability.

In analysing the interaction profiles with hinge region residues (137–141), STK038175 showed a binding pattern similar to the co-crystallized ligand, albeit with nuanced differences (Figure 9a,b, Appendix A), which could explain its superior stability profile during the MD simulation. For example, at Leu137, both ligands displayed comparable van der Waals interactions, but STK038175 exhibited slightly stronger electrostatic attractions and a slight advantage in polar solvation. More pronounced differences were observed at Asp138, where STK038175 and the co-crystallized ligand exhibited similar van der Waals forces, yet their electrostatic interactions diverged, with STK038175 showing potential repulsion in contrast to the attractive forces of the co-crystallized ligand, which also excelled in polar solvation. Furthermore, at Met140, the co-crystallized ligand demonstrated stronger van der Waals and electrostatic interactions, suggesting a higher binding affinity. The positional adjustment of the co-crystallized ligand throughout the simulation may have facilitated these increased interactions with the hinge region residues, enhancing its binding dynamics.

GR04 and STK300222 both demonstrated ligand movement during the simulation, albeit with varying degrees of adjustment. For GR04, the movement was subtle and mirrored the behaviour of the co-crystallized ligand during the simulation. It maintained a consistent conformation throughout the simulation, making only minor adjustments within the active site (Figure 10). Although it was capable of forming interactions with nearby residues such as Val69 and Lys84, its interactions with the hinge region residues remained weak and insufficient. The slight readjustments observed during the simulation did not enhance its interaction profile with the critical hinge region.

Meanwhile, STK300222 undergoes a significant reorientation in the active site, which explains the dramatic fluctuations observed in the RMSD plot up to ~80 ns. Initially, during the first 50 ns, the ligand is positioned in a more horizontal manner within the active site (Figure 11a). However, post-50 ns, it dramatically shifts, pointing its anilino pyrimidinol side chain downwards (Figure 11b). From 100 ns to the end of the 200 ns simulation, STK300222 maintains this orientation with minimal movement, contributing to the stabilization noted in the latter half of the RMSD plot. The interactions at 50 ns are relatively limited, yet this orientation does facilitate at least one interaction with a hinge region residue (Met140). Following its reorientation, STK300222 formed additional interactions with nearby residues including Tyr66, Lys84, and Asp200; however, it did not engage further with the hinge region.

Upon comparison of the pairwise assessment of GR04 and STK300222 (Figure 12a,b, and Appendix A) with the cocrystalized ligand in 5BYZ, it is evident that both GR04 and STK300222 generally exhibit a weaker interaction profile with the hinge region residues compared to the cocrystalized ligand. Although these candidates may show reduced electrostatic repulsion and slightly improved solvation properties in some cases, their overall van der Waals and electrostatic interaction strengths are consistently weaker across most key residues, particularly Met140 and Glu141 for STK300222. This pattern suggests that while they may offer certain solvation advantages, their capacity to form strong, stable interactions with the hinge region of the protein is less favourable than that of 5BYZ, potentially impacting their effectiveness as inhibitors.

### 2.7. ADME Profile of Top Hits

A pharmacokinetic and drug-likeness evaluation was conducted to compare the three top hits against the co-crystallized ligand (Table 2).

All three compounds were predicted to exhibit high gastrointestinal (GI) absorption, similar to 5BYZ. GR04 and STK300222 were predicted to penetrate the blood–brain barrier (BBB), whereas STK038175, like 5BYZ, was not. Furthermore, 5BYZ, STK038175, and GR04 were identified as substrates of P-glycoprotein (P-gp), a membrane-bound efflux transporter abundantly expressed at the BBB. This efflux system plays a protective role by preventing central accumulation of its substrates, thereby reducing the risk of neurotoxicity [29]. Notably, this protective mechanism does not apply to STK300222; thus, medicinal chemistry strategies such as increasing molecular polarity, introducing P-gp recognition motifs, or optimizing lipophilicity may be explored to decrease BBB penetration without compromising target binding [30].

At the same time, multidrug resistance (MDR) may arise because P-gp efflux can lower intracellular drug concentrations and diminish therapeutic efficacy, a limitation reported for several kinase inhibitors currently in clinical use [29]. Accordingly, future optimization of our scaffolds should consider strategies to minimize P-gp efflux liability, such as structural modifications to reduce recognition, co-administration with P-gp inhibitors, or formulation approaches to improve intracellular retention. Importantly, optimization must balance peripheral exposure, efflux liability, and CNS penetration, since excessive reduction in efflux could increase the risk of neurotoxicity.

Unlike 5BYZ, all three selected compounds were also predicted to participate in possible drug–drug interactions, as they demonstrated the potential to bind with at least three distinct cytochrome P450 isoforms. All three of the top hits fulfil the Druglikeness rules, which is not the case for 5BYZ cocrystalized ligand, as it violates the molar refractivity, exceeding the upper limit of 130. They also all appear to follow the Leadlikeness rules, unlike 5BYZ, as its molecular weight exceed 350 Da, and its rotatable bonds exceed 7, making it a highly flexible and sizeable ligand. While our top hits have a profile that needs optimization in terms of not only safety but effectivity as well, they are small and simple enough structures, which would allow for optimization and derivatization.

As the ATP-binding site of ERK5 is highly conserved across other members of the MAPK family, this gives rise to specificity issues, which is one limitation of our current leads. As such, non-specific inhibition remains a potential concern for small molecules designed against the ATP-binding site of ERK5. While our workflow was successful in identifying novel lead-like compounds, the assays performed did not directly assess selectivity over other kinases. Moving forward, structure-based strategies such as exploiting unique features of ERK5—including its extended C-terminal domain and distinct hinge interactions [4,9,31,32] together with kinase panel profiling—will be essential to ensure specificity. Encouragingly, prior studies have demonstrated that subtle differences in hinge residue composition and pocket topology can be leveraged to design inhibitors with improved selectivity toward ERK5 [18]. Our current hits, particularly STK300222 and GR04, are sufficiently small, making them ideal for derivatization aimed at enhancing their ERK5 selectivity in future optimization efforts.

## 3. Materials and Methods

### 3.1. Structure-Based Virtual Screening

#### 3.1.1. Protein (Crystal Structure) Selection and Validation for Docking-Based Virtual Screening

All ERK5 crystal structures (a total of 12 PDBs) were downloaded for evaluation from the protein data bank. Unliganded crystal structures (apo forms) were discarded. Additionally, crystal structures with inhibitors bound in allosteric sites were also discarded. The remaining 8 crystal structures were evaluated to study their co-crystallized ligands. If the housed ligand violated the drug-like rules in terms of size (molecular weight (M.W.) > 500 da), it was discarded. This is because the ATP-binding site would be kept rigid during the docking protocol, and hence through induced fit, it would be best suited for bigger ligands if the co-crystallized ligand was larger in terms of M.W. The remaining 6 PDBs were then aligned on top of one another, to evaluated them in terms of important missing residues and chains. Structures with missing residues near the ATP-binding cavity were eliminated. The remaining 3 PDBs (5BYZ [31], 5O7I [32], 7PUS [4]) free of any mutations, and with co-crystallized ligands that have drug-like properties that are bound to the ATP-binding site were compared in terms of their resolution. As 5BYZ had the best resolution of 1.65 Å (as opposed to 2.38 Å and 2.59 Å in the case of 5O7I and 7PUS, respectively), it was picked for the docking based virtual screening protocol.

As part of the validation procedure preceding the virtual screening workflow, self-docking was carried out to assess the reproducibility of the ligand’s crystallographic binding orientation. The redocked pose generated using XP-docking was aligned with the original structure, yielding an RMSD of 1.26 Å. This result confirms that the docking protocol is capable of accurately reproducing experimentally observed binding conformations within the ERK5 ATP-binding site.

#### 3.1.2. Protein Preparation and Grid Generation

The crystal structure of ERK5 (PDB: 5BYZ [31]) was downloaded from the RCSB PDB. To prepare the structure, all water molecules were initially removed. Subsequently, MOE’s protein preparation tool was used to model missing residues and incorporate hydrogen atoms [33]. Further processing was performed using the Protein Preparation module in Maestro Schrödinger [34]. At this stage, optimization of hydrogen atoms was performed, and refinement of the structure was performed through restrained minimization, achieving a maximum RMSD of 0.30 Å to reach a lower-energy conformation. Generation of receptor grid for docking was performed around the centroid of the ligand found in the crystal structure. To ensure complete coverage of the binding cavity, the grid box was set to 15 Å and 26 Å, for the inner and outer boxes, respectively.

#### 3.1.3. Ligand Preparation

A ligand database was constructed from two commercially available libraries: TimTec [35], and ChemDiv [36], making up a library of ~1.6 million ligands (1,603,576) pre-filtration. The compiled library was filtered as per a number of drug-like rules, namely Lipinski’s [37] and Veber’s [38] rules (molecular weight (M.W.) ≤ 500 Da, LogP < 5, number of hydrogen bond acceptor (HBA) ≤ 10, number of hydrogen bond donor (HBD) ≤ 5, number of rotatable bond (RotB) ≤ 10, total polar surface area (TPSA) ≤ 140). The ligands remaining were then passed through a PAINS-Remover filter [39], to further filter our library removing potentially promiscuous compounds, resulting in a final library of ~1.1 million (1,126,120) ligands. Lastly, the drug-like filtered library was processed using the LigPrep module [40] in Maestro [34].

#### 3.1.4. Docking and Shortlisting of Hits

The promiscuous free, drug-like filtered library was docked into the ATP-binding cavity using the Glide workflow integrated within Maestro Schrödinger [41]. The virtual screening process was carried out in three sequential steps of increasing accuracy: high-throughput virtual screening (HTVS), standard precision (SP), and extra precision (XP). At each stage, approximately 20% of the top-ranking ligands from the previous round were retained and advanced to the next tier of docking refinement.

Following the XP docking step, the top 500 scoring ligands were visually examined to evaluate their binding orientation, focusing on their accommodation within the pocket and interactions with adjacent key residues. Selected molecules were further analysed using aggregator prediction tools such as Aggregator Advisor [42] and BAD Molecule Filter [43]. The use of such filters ensured that no candidate exhibited undesirable behaviours such as nonspecific binding or aggregation. Finally, pharmacokinetic and drug-likeness evaluations were performed using SwissADME [44], and compounds that passed all screening criteria were selected for biological testing in cell-based assays.

### 3.2. Cell Culture

Two lung cancer cell lines were utilized in the present study (H292 and A549). H292 (#91091815) and A549 (#86012804) cell lines were obtained from Sigma, St. Louis, MO, USA. The cells were cultured in RPMI-1640 medium (#R8758, Sigma, USA), supplemented with 10% fetal bovine serum (FBS; #F9665, Sigma, USA) and antibiotics (100 U/mL penicillin and 100 µg/mL streptomycin; #A5955, Sigma, USA). The cultures were maintained in a humidified incubator at 37 °C with 5% CO_2_ to ensure optimal growth conditions.

### 3.3. Cell Proliferation Assay

Colorimetric MTT metabolic activity assay (#ab211091, Abcam, Cambridge, UK) was conducted for the evaluation of cell proliferation. Cancer cell lines were seeded into a 96-well plate at a density of $5\times 10^{4}$ cells/well and incubated at 37 °C. An initial concentration of 100 µM was used for each of the 40 compounds during the primary screening. From this screening, three promising candidates, STK038175, STK300222, and GR04, were selected for further evaluation at varying concentrations (0.1 µM, 1 µM, 10 µM, and 100 µM), along with DMSO (#D2650, Sigma, USA) as a control.

Treatments were applied for 72 h. Following the incubation period, the supernatant was removed, and the cells were washed twice with 1× PBS. Subsequently, 20 µL of MTT solution was added to each well and incubated for 3 h. Absorbance was measured at 570 nm using a spectrophotometer

### 3.4. Wound Healing Assay

Human cancer cell lines ($5\times 10^{4}$ cells/well) were seeded into 96-well plates and treated with one of the following: STK038175 (10 µM or 20 µM), STK300222 (25 µM or 50 µM), GR04 (10 µM or 20 µM), or DMSO (#D2650, Sigma, USA). A linear scratch was created across the cell monolayer using a sterile 200 µL micropipette tip. The plates were incubated at 37 °C in a humidified incubator, and images of the scratched area were captured at 0, 24, 48, and 72 h. The wound closure was analysed using ImageJ software, version 1.54f.

### 3.5. Protein Extraction

Cells were treated with STK038175 (10 µM or 20 µM), STK300222 (25 µM or 50 µM), GR04 (10 µM or 20 µM), XMD8-92 (1234480-50-2, Sigma, USA; 10 µM) [45,46] as a known ERK5 inhibitor, or DMSO (#D2650, Sigma, USA; 10 µM) for 72 h. Upon reaching confluency, live cells were lysed directly in a buffer containing 150 mM NaCl, 0.1% sodium dodecyl sulphate (SDS), 1% Nonidet P40, 0.5% sodium deoxycholate, potassium dihydrogen phosphate, sodium phosphate anhydrous dibasic, and Complete Protease Inhibitors. The lysates were centrifuged at 12,000 rpm for 30 min, and the supernatant was collected as the cell extract. Protein concentrations were quantified using the BCA Protein Assay Kit (#23225, Thermo Fisher Scientific, Waltham, MA, USA) with BSA as the standard.

### 3.6. SDS-PAGE and Western Blot

Forty micrograms of total protein were loaded per lane onto SDS-polyacrylamide gels for electrophoresis (PAGE). The separated proteins were transferred to nitrocellulose membranes, which were then blocked for 1 h with 5% BSA prepared in Tween-TBS (10 mM Tris-HCl, pH 7.5; 100 mM NaCl; 0.1% Tween-20).

The membranes were incubated overnight at 4 °C with primary antibodies at a 1:700 dilution, including Phospho-c-Fos (#5348, Cell Signalling, Danvers, MA, USA), Phospho-FRA1 (#5841, Cell Signalling, USA), Phospho-CREB (#9198, Cell Signalling, USA), and GAPDH (#2118, Cell Signalling, USA). After three washes with Tween-TBS, membranes were treated with an anti-rabbit IgG HRP-linked secondary antibody (#7074, Cell Signalling, USA) at a 1:2000 dilution for 1 h. Proteins were visualized using HRP-conjugated secondary antibodies, and quantification was performed with ImageJ software.

### 3.7. Molecular Dynamics Simulations

#### 3.7.1. MD Simulation, MM-GB/SA Rescoring and Pairwise Residue Analysis

Molecular dynamics (MD) simulations were performed on the three lead compounds STK038175, STK300222, and GR04, to investigate their stability within the designated ATP-binding site. MD was carried out using the AMBER18 software package [47]. Partial charges as well as other molecular protein parameters were assigned to the protein using the ff19SB force field. Ligand parameters were defined using the generalized AMBER force field (GAFF). The system was built using xleap in AmberTools and subsequently neutralized with three Na^+^ counter ions. Solvation of the system was performed in a TIP3P water box, producing a truncated octahedral solvation box. Energy minimization was then performed in two subsequent steps using the PMEMD engine in AMBER18. The first phase involved restraining all non-solvent atoms (solute atoms) with a force constant of 500 Kcal mol^−1^Å^−1^ for 1000 cycles, followed by a second unrestrained minimization for an additional 1000 cycles.

The system underwent stepwise heating to reach 300 K under the NVT ensemble, with hydrogen-involved bond lengths constrained via the SHAKE algorithm and temperature regulated using the Langevin thermostat at a collision frequency of 1.0 ps^−1^. A 200 ns MD production run followed under the NPT ensemble, keeping the temperature and pressure constant at 300 K and 1.01 × 10^5^ Pa, respectively. Binding energies were subsequently evaluated using MM-GB/SA [48]. The analysis was limited to the final 30 ns of the trajectory to confirm ligand stabilization while minimizing the effect of any transient conformational convergences within the binding site.

To evaluate the energetic contributions of individual ligand–residue interactions within the binding site, pairwise decomposition analysis (idecomp = 4) was applied to the MD simulation trajectories [49].

#### 3.7.2. Clustering Analysis of Produced MD Trajectories

Trajectory analysis was initiated following the completion of MD simulations, beginning with the removal of all solvent molecules and ions. Frames were then sampled every 10 ns to enable clustering, which was performed using the DBSCAN algorithm [50] implemented via the cpptraj module in AmberTools [51]. A default epsilon (ε) value of 3.0 was retained to define the maximum distance threshold for cluster formation. RMSD plots were subsequently generated from the resulting trajectory files to assist in evaluating conformational stability over time.

### 3.8. Statistical Analysis

Results are expressed as the mean ± standard deviation, calculated using three independent replicates. Differences between groups were evaluated through two-way ANOVA using GraphPad Prism version 9.5.1 (GraphPad Software, Boston, MA, USA). A *p*-value below 0.05 was considered indicative of statistical significance. Asterisks in the figures denote levels of significance: * for *p* < 0.05, ** for *p* < 0.01, and *** for *p* < 0.001.

## 4. Conclusions

This study successfully identified novel small molecule inhibitors of ERK5, a kinase integral to cell proliferation and survival pathways, particularly relevant in the context of lung cancer. Through a rigorous process of structure-based virtual screening, biological evaluation, and molecular dynamics simulations, we identified three promising compounds—STK038175, STK300222, and GR04—with significant potential to inhibit ERK5 activity.

Among these, STK300222 emerged as particularly promising due to its substantial inhibitory effect on phosphorylation of ERK5 down-stream targets and its marked impact on reducing cancer cell migration in wound healing assays. This compound not only demonstrated efficacy comparable to the known ERK5 inhibitor XMD8-92 but also showed a favourable stability profile in molecular dynamics simulations, suggesting a robust interaction with ERK5. GR04 also displayed significant antiproliferative activity and, along with STK300222, warrants further optimization to enhance potency and specificity.

The pharmacokinetic properties predicted by SwissADME indicate high gastrointestinal absorption for all three hits, with GR04 and STK300222 predicted to pass the blood–brain barrier. The compounds were generally predicted to interact with multiple cytochrome P450 enzymes, indicating potential drug–drug interactions. These ADME profiles highlight the necessity for further optimization of these inhibitors to improve safety and efficacy.

Further in vivo studies and comprehensive pharmacokinetic profiling are required to confirm the therapeutic potential and safety of these compounds, which could eventually lead to novel treatments for cancer patients targeting the ERK5 signalling pathway. The detailed molecular analysis revealed the interactions and stability of the lead compounds within the ERK5 binding site, providing valuable insights into the binding dynamics that can guide further structural optimization.

## Figures and Tables

**Figure 1 molecules-30-04181-f001:**
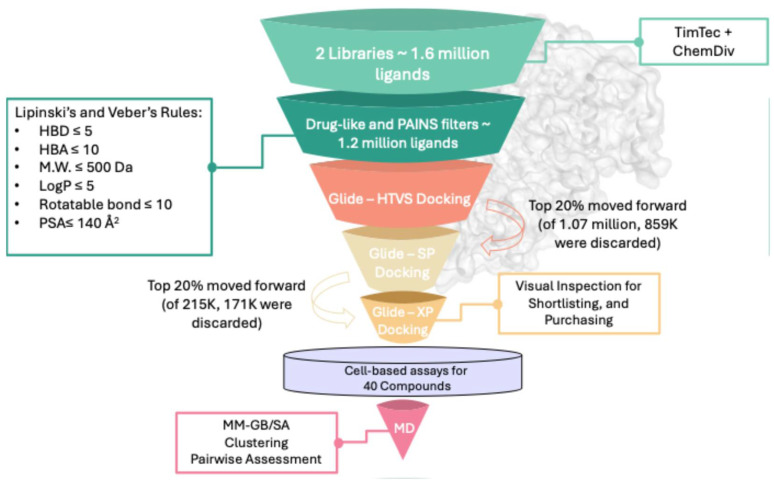
Docking-based virtual screening conducted against the ATP-binding site of ERK5.

**Figure 2 molecules-30-04181-f002:**
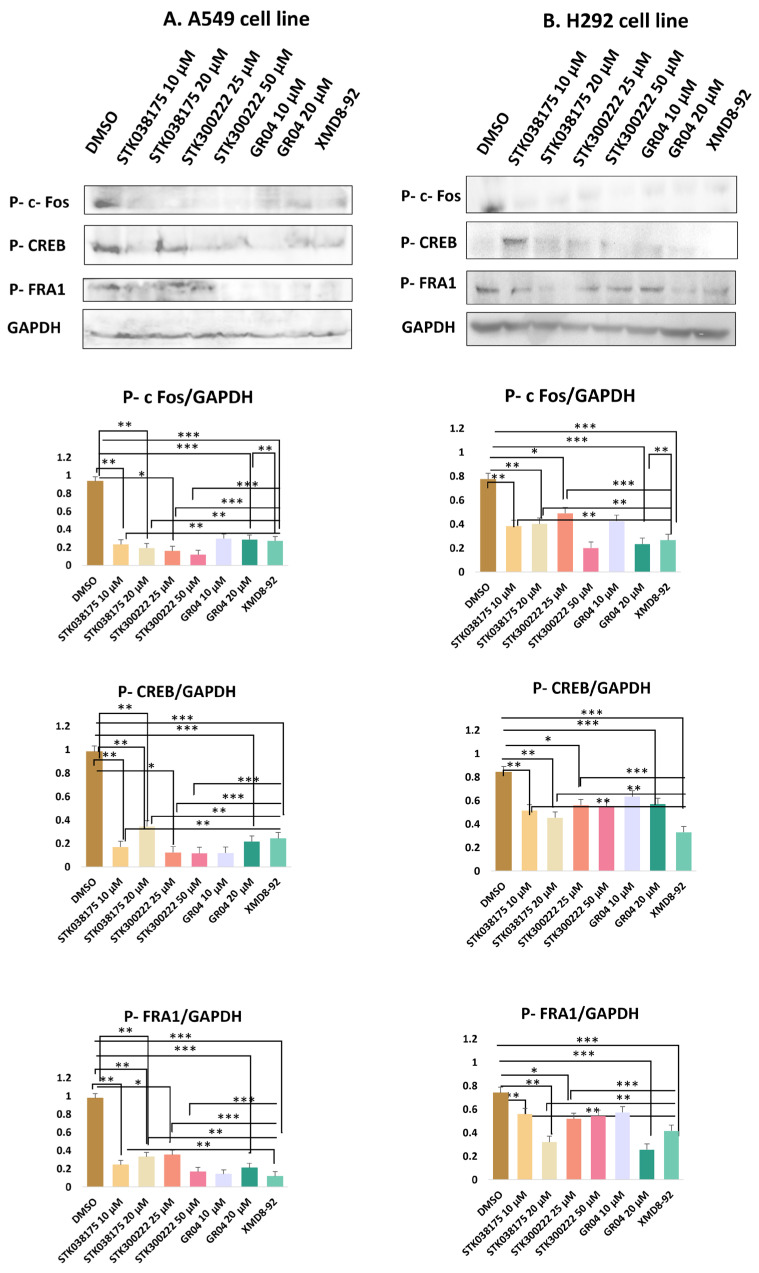
Western blot analysis of ERK5-dependent phosphorylation in (**A**) A549 and (**B**) H292 lung cancer cell lines treated with potential ERK5 inhibitors STK03817 (10, 20 µM), STK300222 (25, 50 µM), GR04 (25, 50 µM), and the reference inhibitor XMD8-92. Phosphorylation of c-Fos, CREB, and Fra1 was assessed, with GAPDH as loading control. Bar graphs show band intensities normalized to GAPDH and expressed relative to DMSO control. Data are mean ± SD (*n* = 3); significance: * *p* < 0.05, ** *p* < 0.01, *** *p* < 0.001 vs. DMSO.

**Figure 3 molecules-30-04181-f003:**
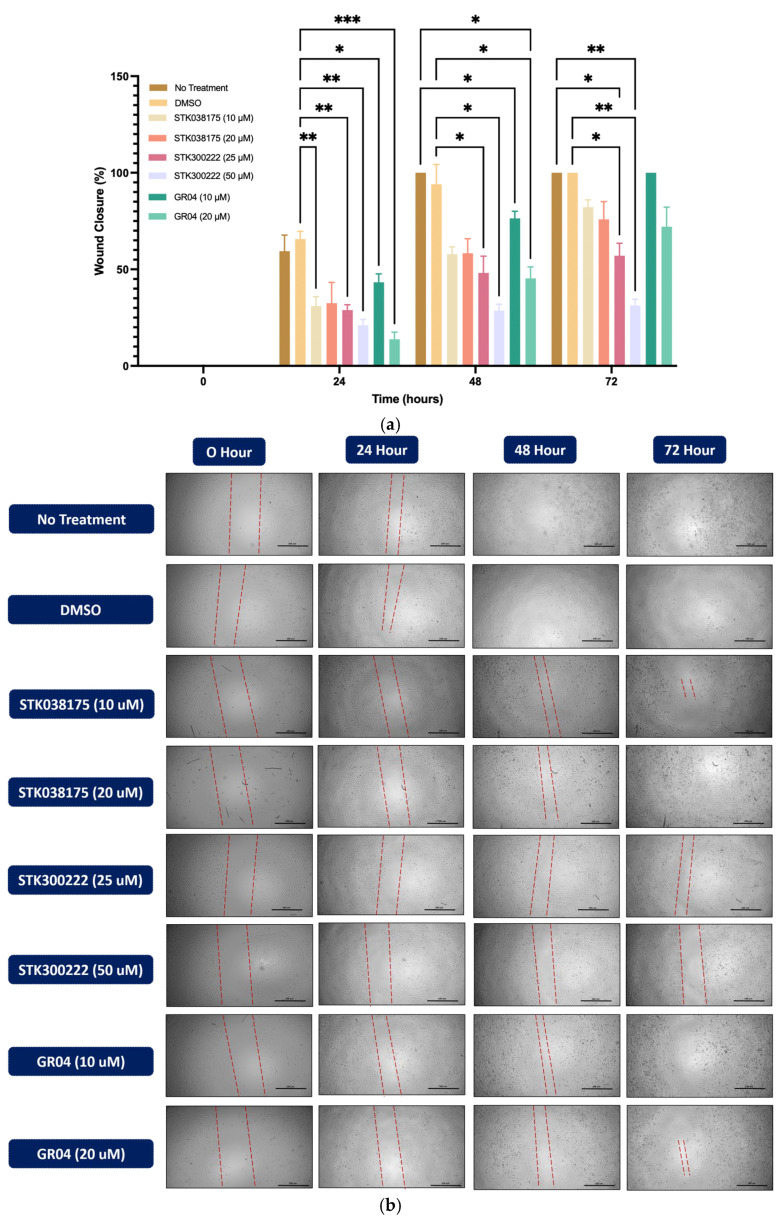
Wound healing assay in A549 lung cancer cells treated with potential ERK5 inhibitors. Cells were exposed to DMSO (control), STK038175 (10, 20 µM), STK300222 (25, 50 µM), GR04 (10, 20 µM), or left untreated. (**a**) Wound closure was quantified at 0, 24, 48, and 72 h, with results shown as mean ± SD (n = 3); * *p* < 0.05, ** *p* < 0.01, *** *p* < 0.001 vs. control. (**b**) Representative images of wound closure at each time point are shown for each condition (4× objective; scale bar = 100 µm).

**Figure 4 molecules-30-04181-f004:**
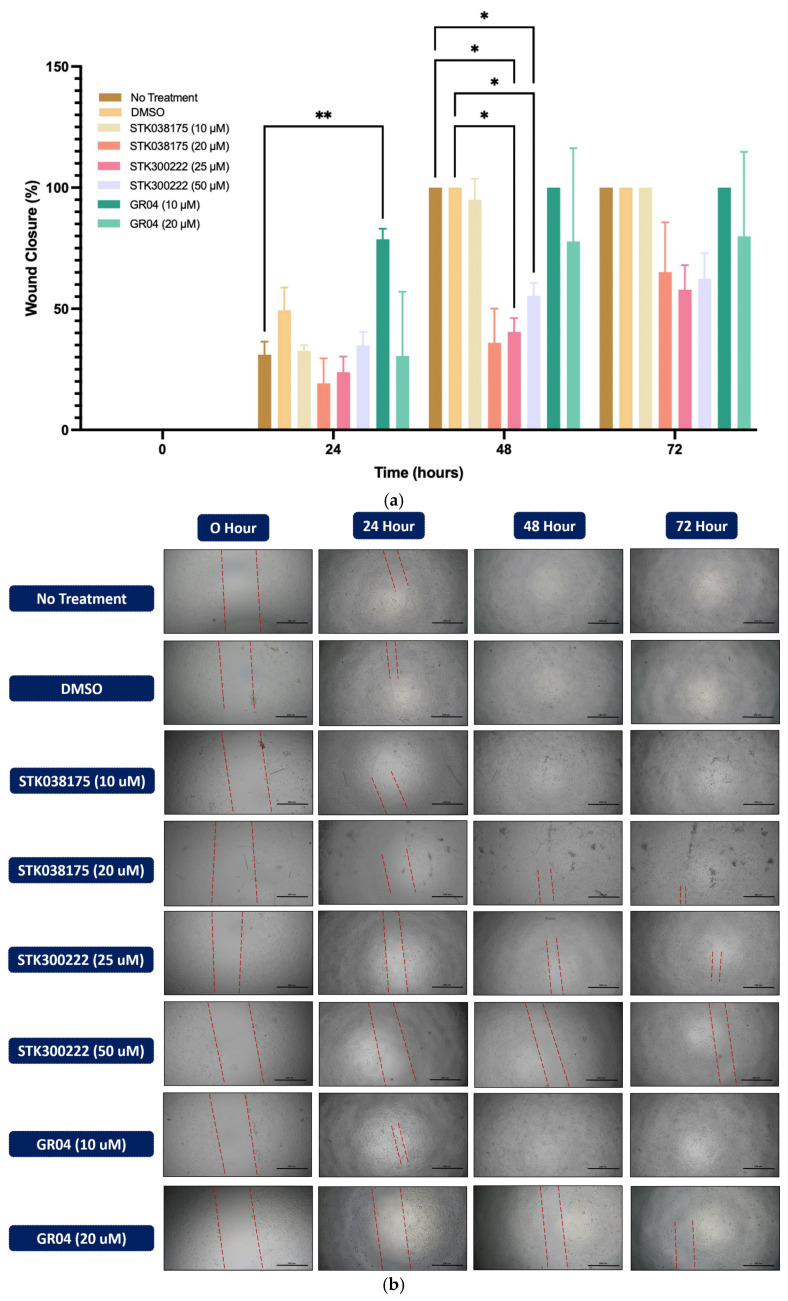
Wound healing assay in H292 lung cancer cells treated with potential ERK5 inhibitors. Cells were treated with DMSO (control), STK038175 (10, 20 µM), STK300222 (25, 50 µM), GR04 (10, 20 µM), or left untreated. (**a**) Wound closure was quantified at 0, 24, 48, and 72 h, shown as mean ± SD (n = 3); * *p* < 0.05, ** *p* < 0.01 vs. control. (**b**) Representative images of wound closure at each time point are shown for each condition (4× objective; scale bar = 100 µm).

**Figure 5 molecules-30-04181-f005:**
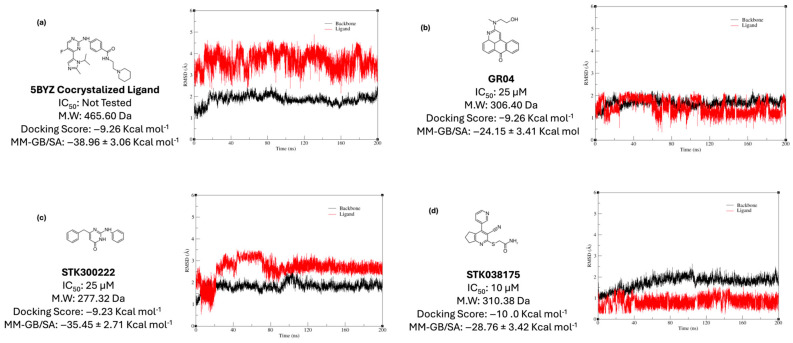
RMSD plots of: (**a**) 5BYZ cocrystalized ligand; (**b**) GR04; (**c**) STK300222, and (**d**) STK038175 across 200 ns MD simulation.

**Figure 6 molecules-30-04181-f006:**
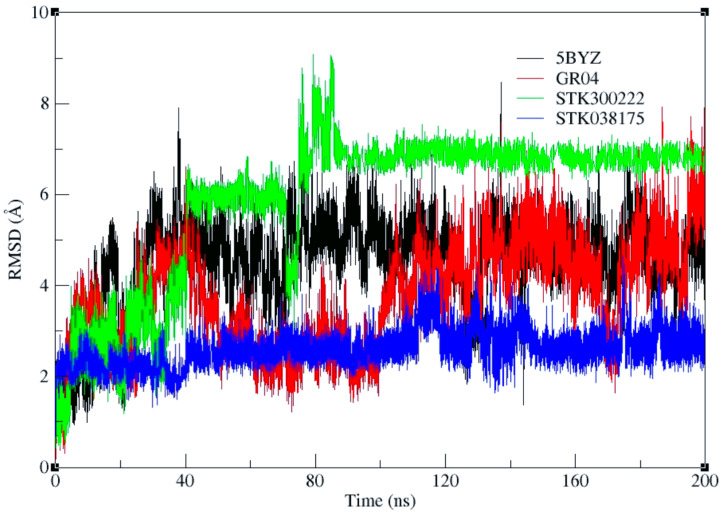
Ligand-to-pocket RMSD plots of 5BYZ cocrystalized ligand, GR04, STK300222, and STK038175 across 200 ns MD simulation.

**Figure 7 molecules-30-04181-f007:**
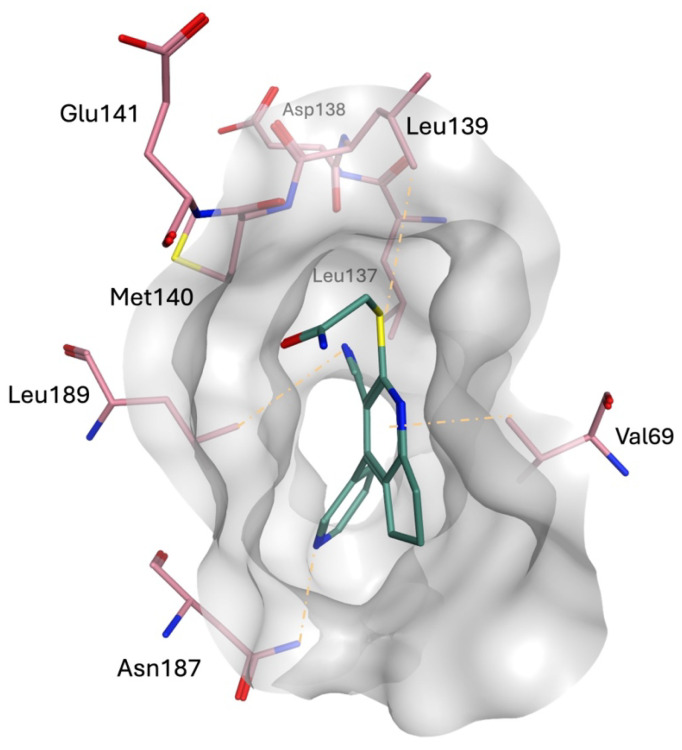
Binding pose and ligand interactions of STK038175 (green sticks) in the ERK5 active site (pink sticks) retrieved from the 200 ns MD simulation.

**Figure 8 molecules-30-04181-f008:**
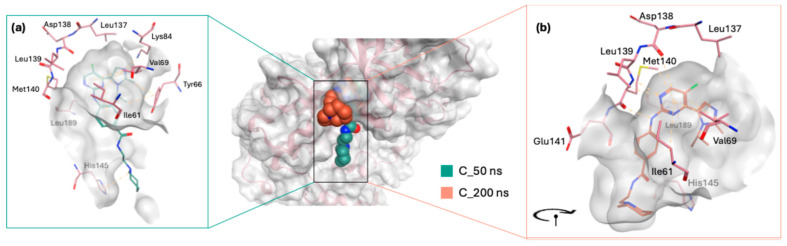
ERK5 in complex with the cocrystalized ligand of 5BYZ during the 200 ns MD simulation. (**a**) Binding mode and ligand interactions of cocrystalized ligand up to 50 ns, and (**b**) up to 200 ns.

**Figure 9 molecules-30-04181-f009:**
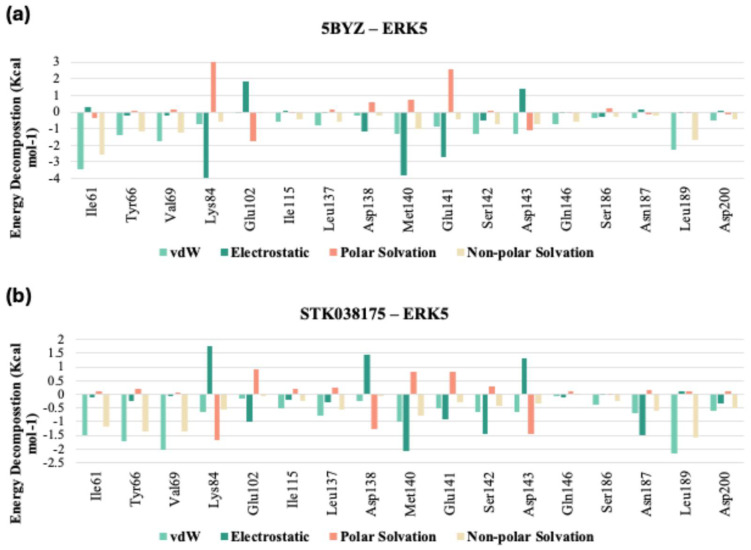
Pairwise Assessment of (**a**) 5BYZ cocrystalized ligand and (**b**) STK038175 from the 200 ns MD trajectories.

**Figure 10 molecules-30-04181-f010:**
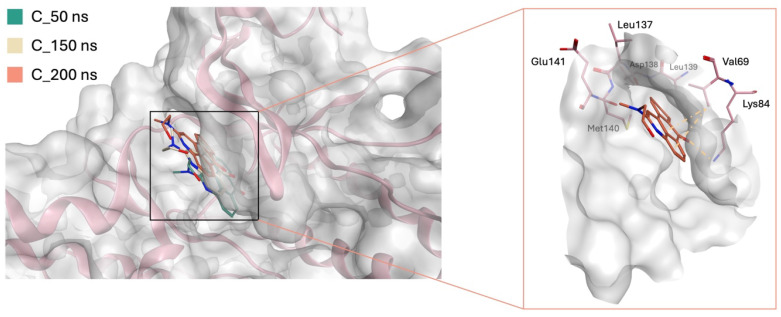
Binding poses and ligand interactions of GR04 in the ERK5 active site retrieved from the 200 ns MD simulation.

**Figure 11 molecules-30-04181-f011:**
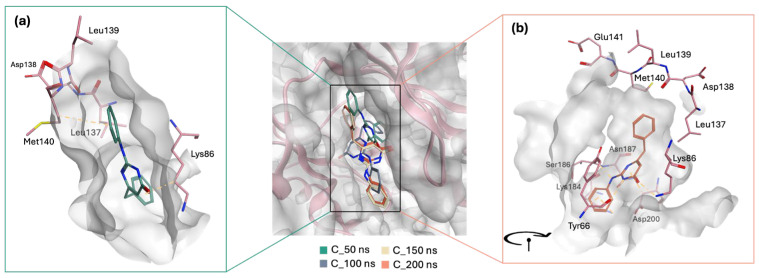
ERK5 in complex with STK300222 during the 200 ns MD simulation. (**a**) Binding mode and ligand interactions of STK300222 at 50 ns, and (**b**) at 200 ns.

**Figure 12 molecules-30-04181-f012:**
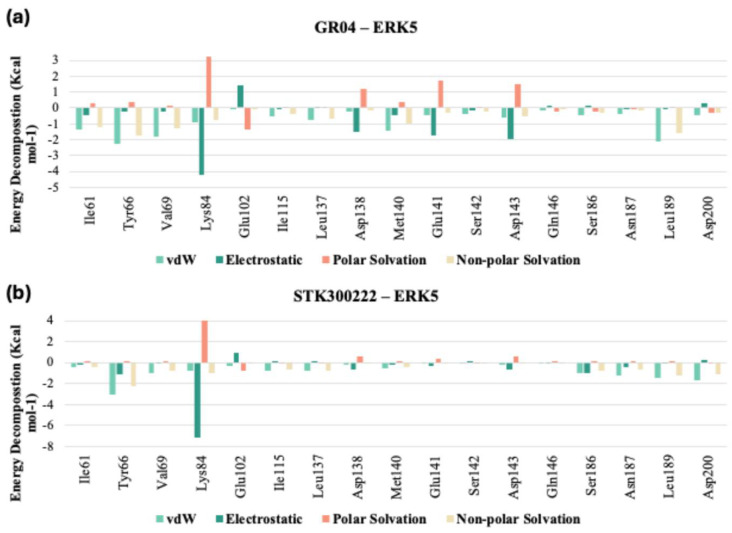
Pairwise Assessment of (**a**) GR04 and (**b**) STK300222 from the 200 ns MD trajectories.

**Table 1 molecules-30-04181-t001:** IC_50_ values and Treatment concentrations used in the experiment for the compounds STK038175, STK300222 and GR04.

Compound ID	Structure	IC_50_ (µM) ± SD	Treatment Concentration 1 (IC_50_)	Treatment Concentration 2(IC_50_ × 2)
STK038175	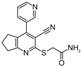	10 ± 0.2	10	20
STK300222	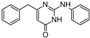	25 ± 0.8	25	50
GR04	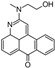	10 ± 0.5	10	20

**Table 2 molecules-30-04181-t002:** Pharmacokinetics and Druglike and Leadlike Characteristics of Selected Hits Predicted by SwissADME.

		5BYZ	STK038175	GR04	STK300222
Pharmacokinetics	GI absorption	High	High	High	High
	BBB permeant	No	No	Yes	Yes
	P-gp substrate	Yes	Yes	Yes	No
	CYP1A2 inhibitor	No	Yes	Yes	Yes
	CYP2C19 inhibitor	No	No	No	Yes
	CYP2C9 inhibitor	No	Yes	Yes	No
	CYP2D6 inhibitor	No	No	Yes	Yes
	CYP3A4 inhibitor	No	Yes	Yes	No
	Bioavailability score	0.55	0.55	0.55	0.55
Druglikeness	Lipinski	Yes	Yes	Yes	Yes
	Ghose	No MR > 130	Yes	Yes	Yes
	Veber	Yes	Yes	Yes	Yes
	Egan	Yes	Yes	Yes	Yes
	Muegge	Yes	Yes	Yes	Yes
Medicinal Chemistry	PAINS	0	0	0	0
	Brenk	0	0	0	0
	Leadlikeness	No MW > 350 Rotors > 7	Yes	Yes	Yes

## Data Availability

The authors confirm that the data supporting the findings of this study are available within the article and its Appendix A. Additional data are available upon reasonable request from the corresponding author.

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
