# Peer review of "The Discovery of Small ERK5 Inhibitors via Structure-Based Virtual Screening, Biological Evaluation and MD Simulations"

_molecules, 2025, doi:10.3390/molecules30214181_

Round 1

Reviewer 1 Report

Comments and Suggestions for Authors

Please see the attached PDF file.

Reviewer 2 Report

Comments and Suggestions for Authors

The comment is attached in word file that need to be addressed

Comments on the Quality of English Language

I cannot evaluate the English but it is understandable.

Reviewer 3 Report

Comments and Suggestions for Authors

This manuscript presents efforts using CADD tools and wet lab assays to search for small molecular binders targeting ERK5. The following comments are suggested for authors' consideration. 

For Fig. 5, it's better to set a consistent Y-axis scale for all sub-plots so that the readers can easily grab information from the figure. 

There seems to be inconsistencies between MM/GBSA predicted binding free energies vs experimental IC50s. MM/GBSA predicts STK300222 is the strongest binder among the three hits while the IC50 ranking is opposite. Could the authors address limitation of the modeling method and possible cause of such discrepancy?  And did Glide docking score rank these compounds correctly?

The template sentences should be removed from section 2: "2. Results and Discussion This section may be divided by subheadings. It should provide a concise and precise 
description of the experimental results, their interpretation, as well as the experimental 
conclusions that can be drawn.
"

For the methods section, sub-section 3.7.3 is actually highly related to the ending part of 3.7.1 since the pairwise residue analysis is actually part of the MM/GBSA analysis and could be combined in some way.

Reviewer 4 Report

Comments and Suggestions for Authors

The authors identified three ERK5 inhibitors using virtual screening protocol followed by cellular assays including MTT assay and MD simulations study. The study is interesting and have high impact, but needs revision to improve the quality of the manuscript. Here are my comments:

  1. All biological validation is cell-based (MTT proliferation, Western blot phosphorylation, and wound healing migration assays). A direct biochemical ERK5 inhibition assay should be included in the manuscript. Without this, it is not possible to confirm that the observed effects are truly ERK5-mediated rather than off-target toxicity.
  2. The MD simulations were performed only once for each compound. A single trajectory is not sufficient to make strong conclusions. These should be repeated in triplicate, and the authors should report averaged RMSD, RMSF, and MM-GBSA binding energies with standard deviations.
  3. The RMSD plots presented for ligands are not clearly explained. It is not stated whether ligand RMSD was calculated relative to the protein backbone (to evaluate diffusion from the binding pocket) or ligand-fit-to-ligand (to assess internal conformational changes). Both analyses are informative in different ways. The authors should clarify the method used and should present both for clarity and accuracy of the calculations.
  4. ICâ‚…â‚€ values are reported in the Table 1 for the three active compounds, but the known ERK5 inhibitor XMD8-92 as a positive control is not included in the same assay. Its ICâ‚…â‚€ should be reported alongside the new compounds for comparison purpose. 
  5. The authors mention testing 40 compounds in the MTT screen, yet only three are shown. All the remaining tested compounds should be reported in supplementary material, including their structures and ICâ‚…â‚€/activity values.
  6. SwissADME predictions show that GR04 and STK300222 are likely BBB permeant. This could be a liability for systemic cancer therapy. The authors should acknowledge this limitation and discuss possible strategies for optimization.
  7. Since identified compounds showed modest ICâ‚…â‚€ values, the possibility of off-target effects on ERK1/2 or other kinases should be addressed because ERK5 shares significant sequence homology with ERK1/2.
  8. Please also provide the number of molecules excluded at each virtual filtering step. This will helps evaluate selection robustness.

Round 2

Reviewer 4 Report

Comments and Suggestions for Authors

Thank you for revising the manuscript according to my suggestions and comments. The authors have addressed most of my points thoroughly; however, the RMSD plots still need clarification. The ligand RMSD relative to the protein backbone must be included in addition to the ligand-fit-to-ligand RMSD, as this is essential to evaluate whether ligands remain in the binding pocket. This analysis is standard and should not take much time.

I also have a few minor edits to further improve the manuscript quality:

  1. Ensure all P values are italicized (e.g., P < 0.05).
  2. Use the proper minus sign (−) instead of a hyphen throughout the manuscript for docking scores/energy scores. 
  3. In Figure 1, add missing units in the Lipinski’s and Veber’s rules section (e.g., Da, Ų).

Author Response

Comment 1: The ligand RMSD relative to the protein backbone must be included in addition to the ligand-fit-to-ligand RMSD, as this is essential to evaluate whether ligands remain in the binding pocket.

Response 1: The authors would like to thank the reviewer for allowing us to revise our manuscript. The additional RMSD calculation was performed and added on Page 12, Lines 334 - 357:

"Besides the original analysis calculating the ligand RMSD relative to the ligand itself (ligand-to-ligand RMSD), primarily reflecting the internal flexibility and conformational changes of the ligand throughout the trajectory, a ligand-to-pocket RMSD analysis was performed to better assess ligand retention within the binding site by aligning the trajectory on the backbone atoms of the binding pocket residues (Figure 6).

The ligand-to-pocket RMSD results reveal distinct behaviours among the four ligands. STK300222 exhibits the highest values, plateauing around 6–7 Å with a brief spike to 8 Å. 5BYZ, the co-crystallized ligand, shows values between 4 and 6.5 Å, while GR04 fluctuates between approximately 3 and 5.5 Å. STK038175 remains the most stable, with RMSD values around 2–3 Å. Visual inspection of the trajectories confirms that none of the ligands dissociate from the binding pocket during the 200 ns simulation. Clustering of the trajectories every 50 ns indicates that the higher RMSD values observed for 5BYZ and STK300222 arise from pose reorientation within the pocket, whereas GR04 shows moderate reorientation and STK038175 remains largely fixed.

Such elevated RMSD values in the absence of dissociation have been previously reported in multiple molecular dynamics studies. One study showed that a ligand bound to CDK2 (PDB 2B52) exhibited RMSD values exceeding 6 Å due to pose rearrangements within the pocket rather than unbinding [25]. Another observed similar behaviour for Abl–imatinib complexes, where high ligand RMSD values occurred despite the ligand remaining stably anchored [26].

Accordingly, our higher RMSDs—especially for 5BYZ and STK300222—are best explained by pose flips/reorientations in the binding pocket, rather than exit from the site."

Comment 2: 

  1. Ensure all P values are italicized (e.g., P < 0.05).
  2. Use the proper minus sign (−) instead of a hyphen throughout the manuscript for docking scores/energy scores. 
  3. In Figure 1, add missing units in the Lipinski’s and Veber’s rules section (e.g., Da, Ų).

Response 2: all three points were edited in the manuscript according to the reviewer's comment.
